# Assessment of Viscoelastic Parameters of Muscles in Children Aged 4–9 Months with Minor Qualitative Impairment of the Motor Pattern after Vojta Therapy Implementation

**DOI:** 10.3390/ijerph191610448

**Published:** 2022-08-22

**Authors:** Agnieszka Ptak, Agnieszka Dębiec-Bąk, Małgorzata Stefańska

**Affiliations:** Faculty of Physiotherapy, Wrocław University of Health and Sport Sciences, al. Ignacego Jana Paderewskiego 35, 51-612 Wrocław, Poland

**Keywords:** viscoelastic properties of the muscle, development, function

## Abstract

The aim of this study was to assess if there are any objective changes in the viscoelastic parameters of the erector spinae muscle after Vojta stimulation. The study involved 22 healthy children at an average age of 7 months and with an Apgar score of 8–10 points, who were referred for rehabilitation due to a slight delay in the phases of psychomotor development. The first group consisted of 11 children with increased muscle tone (IMT) and the second group consisted of 11 children with non-increased muscle tone (nonIMT). All study participants received a one-time Vojta therapy session, which was continued for 4 weeks by parents at home. The viscoelastic parameters of the dorsal extensor muscle were measured three times. In the first study group, changes in the viscoelastic parameters of the extensor muscles of the back occurred immediately after the therapy at the first examination, whereas changes in the supporting and extensor function of the limbs occurred in both groups at the second examination. Analysis featuring an objective assessment allows physiotherapists to diagnose local changes in the viscoelastic parameters after the implementation of therapy. These studies are the first pilot studies to be continued with a 30- or 60-day follow-up.

## 1. Introduction

It is universally accepted that healthy neonates and infants grow in a specific manner according to developmental norms. The authors are in consensus about the intervals and order of emergence of new skills during the first year of life, referred to as developmental milestones [1]. In these stages, the only factors that cause any developmental discrepancies between children are the welfare and social circumstances in which the children are raised [2]. Even healthy children who received an Apgar score of 8–10 may experience impaired motor development. In the 10th revision of the International Classification of Diseases (ICD-10), such abnormalities are defined as “Lack of expected normal physiological development, unspecified” (code R62.9). Because paediatricians did not find any other abnormalities in the children (only a delay in motor development) the children were referred to a physiotherapist with the diagnosis R62.9. Paediatricians further diagnosed impairments such as abnormal muscle tone and the absence of support and extension function (deficit of motor functions appropriate for the given age), accompanied by food intake difficulties and gastrointestinal issues. In neonates, gastrointestinal problems occur in the form of discoordination of sucking, breathing, and swallowing. Signs of these abnormalities include, pulling away while being breastfed, adopting an improper extension posture, raising shoulders while being breastfed, excessive spitting up, and general restlessness [3,4,5]. These issues can be observed during the assessment of spontaneous motor skills [6,7]. One of the causes of postural abnormalities and discoordination of breathing, swallowing, and sucking, as well as minor issues referred to as qualitative motor function abnormalities, is the dysfunction of deep muscles [8,9].

Under the right conditions, trunk stability refers to the ability of the core, located at the centre of gravity of the body, to maintain or regulate body conditions according to the changes in the external environment [10,11]. In particular, since the trunk muscles play the role of a corset that stabilises the body and the spine, whether the extremities move or not, they are critically important for the maintenance of postures [8,12].

In addition, the trunk muscles maintain body alignment during sitting and standing postures, and act as supports for performing various functions [13,14].

If the function is limited, the performance of postural muscles is impaired. These muscles play a critical role in achieving the stabilisation of the entire body as the starting point for subsequent developmental milestones. One of the methods for conducting a qualitative assessment and therapy of spontaneous motor skills is the Vojta approach [15,16].

The basic principle of Vojta reflex locomotion is the maintenance of postures through isometric contraction of muscles during point (breast zone) stimulation, thereby ensuring consistent patterns of muscle contraction and leading to the stimulation of muscles, joints, ligaments, and tendons. In addition, Vojta reflex locomotion is known to be related to the exteroceptors and the interoceptors, and to act as a source of afferent stimulation going into the central nervous system [17]. Vojta reflex locomotion has been reported to activate the trunk muscles and the deep muscles of the spine to regulate trunk stability and increase spinal rotation force, thereby enhancing postural control [17].

Assessing muscle function in infants is difficult because it is impossible to establish logical contact with the patient. Methods that require active participation in the assessment cannot be followed. Myotonometry is a method of measurement that does not require the patient to actively participate in the assessment process. It allows physiotherapists to estimate various tissue characteristics, such as stiffness and elasticity, and, consequently, to indirectly assess the viscoelastic parameters of the examined muscle in children, as well as any changes over the course of therapy [18,19,20,21].

The aim of this study was to assess if there are any objective changes in the viscoelastic parameters of the erector spinae muscle after Vojta stimulation.

## 2. Materials and Methods

### 2.1. Study Group

The study group consisted of 22 healthy infants referred by a paediatrician to a physiotherapist due to minor impairment of spontaneous motor skills, diagnosed further as abnormal muscle tone, observed during postnatal check-up visits (between the 4th and 6th week of life). Based on the medical diagnosis, the children were divided into two groups. The increased muscle tone group (IMT group) consisted of 11 children with increased general muscle tone. The non-increased muscle tone group (nonIMT group) consisted of 11 children with non-increased general muscle tone. In both groups, food intake difficulties were reported during the taking of medical history. During the assessment of the spontaneous motor skills, all infants were found to have a reclined head position and the support and extension function of the upper extremities was absent or inadequate for their age. Furthermore, significant pelvic anteversion was observed in the IMT group.

The detailed characteristics of both study groups are presented in Table 1.

The inclusion criteria included good general health, an Apgar score of 8–10, and the presence of minor qualitative impairment of spontaneous motor skills as diagnosed by a paediatrician. The exclusion criteria included postural asymmetry, prematurity, as well as neurological and genetic deficits and disorders.

The process of qualification for the research is presented in Figure 1.

The parents of all included children were informed of the manner of execution and the purpose of the study. They gave written consent to their children’s participation in the study and the anonymous publication of the results. The project was approved by the Commission for Scientific Research of the Wrocław University of Health and Sport Sciences, approval No. 24/2021. Trial registration: The research was conducted as a pilot study for the scientific project approved by the Commission for Scientific Research of the Wrocław University of Health and Sport Sciences No. 24/2021. The study is registered in the Australian New Zealand Clinical Trials Registry under the number ACTRN12622000417785, date of registration 11 March 2022.

### 2.2. Research Methods

The children underwent a physiotherapeutic assessment carried out by an experienced paediatric physiotherapist. The normalisation of the distribution of muscle tone was indirectly assessed through the evaluation of the development of spontaneous motor skills by the development tables compiled by the Munich Functional Developmental Diagnostic (MFDD) [17,22,23,24]. Due to the fact that all study participants exhibited minor impairment of spontaneous motor skills, consisting in the support and extension function being absent or inadequate for their age, the children underwent a one-time therapy session that included the stimulation of the breast zone—phase 1 of reflex rolling—according to the Vojta approach [17]. Furthermore, the parents were appropriately instructed and advised to continue the stimulation at home, four times daily until the follow-up visit, which took place 4 weeks later.

The viscoelastic parameters of muscles were measured three times—immediately before stimulation, after stimulation, and during the follow-up visit 4 weeks later. The MYOTON device by Myoton AS (Estonia) was used, which allowed the physiotherapist to assess superficial muscle groups by way of recording the damped natural oscillation of soft biological tissue in the form of an acceleration signal and the subsequent simultaneous computation of the parameters of the state of tension, biomechanical, and viscoelastic properties. Damped natural oscillation is induced by an exterior, low force, quick-release mechanical impulse under constant pre-load. The assessment allowed the physiotherapist to analyse the following parameters: oscillation frequency (Hz), which characterises the intrinsic tension of muscles at the cellular level, and stiffness (N/m), which describe the muscle’s resistance to an external force that deforms its initial shape; elasticity, which defines the muscle’s ability to recover its initial shape after a contraction or removal of an external deforming force; relaxation (ms), which is the time the muscle needs to recover its initial shape after deformation caused by a voluntary contraction or an external force; and creep, which is a ratio of deformation and relaxation time.

In both study groups, the measurements concerned the erector spinae muscle and were taken at the thoracolumbar transition zone (Th-L). The children were examined in a lateral recumbent position in their mothers’ arms, which allowed muscle relaxation and resting measurement. Each result was recorded based on three measurements. The mean value of three measurements for each parameter was used during further analysis.

### 2.3. Statistical Analysis

The Shapiro-Wilk test was used to check the distribution of all measured variables. In the majority of cases, the distribution was found to be close to normal. In both study groups, descriptive statistics (mean values, median, and standard deviation) were calculated for each measurement. The independent samples *t*-test was used to compare the values obtained in both groups (homogeneity of variance, measured using the Levene’s test, was maintained). Repeated measures ANOVA was used to analyse the differences between examinations 1, 2, and 3. Where the analysis of variance showed significant differences between examinations, a post hoc Scheffe’s test was used. The analysis was carried out in Statistica 13.3.

## 3. Results

The myotonometric measurement results consisted of the values of frequency, stiffness, elasticity, relaxation, and creep, obtained from each of the three measurements of the erector spinae. Differences between the values obtained in both study groups were observed (Table 2). However, only the differences observed during the second measurement (immediately after stimulation) were statistically significant (Table 3).

The change in the myotonometric measurement result under the influence of the stimulus, i.e., a one-time therapy session according to the Vojta approach, was analysed. Various responses to stimulation were observed among the children. In the group of children with non-increased muscle tone (nonIMT group), there were no significant differences between the measurements taken before the stimulation, immediately after stimulation, and after 4 weeks. In the group of children with increased muscle tone, the authors observed statistically significant differences in the frequency of muscle impulses, muscle stiffness, and—during the rest between measurements 1 and 2 (taken before and after stimulation) and between measurements 2 and 3 (taken immediately after the stimulus and after 4 weeks)—also in the frequency and resting time (Table 4).

## 4. Discussion

Healthy, full-term children, i.e., born between the 38th and 42nd week of gestation, with an Apgar score of 8–10, exhibit the development of motor patterns which emerge sequentially every 6 weeks. Those patterns have both qualitative and quantitative aspects [2,15,18]. Minor deficits of spontaneous motor skills manifest as limited or absent support and extension function of the upper extremities in the prone position, accompanied by food intake difficulties and digestive issues [6,7,19]. Minor limitations concerning spontaneous motor skills are defined as qualitative impairment of the motor pattern. They may cause difficulty in nursing (sucking) and impact the support and extension function of the upper extremities in neonates and infants [25].

In this study, in the initial examination, the children with increased and non-elevated muscle tone did not differ in a statistically significant manner, while after the therapy they differed in the frequency, stiffness, relaxation, and creep parameters. In response to the therapy, statistically significant changes were observed only in the group of children with increased muscle tone.

Therapy according to the Vojta approach allows physiotherapists to carry out a subjective assessment of the motor pattern by comparing the pattern exhibited by the child with the reference standard described in the literature [24]. Through the activation of postural muscles, the therapy influences the qualitative aspect of motor development in neonates, infants, and children. The appropriate quality of the motor pattern is crucial for the healthy postural development of the child, marked by the acquisition of consecutive motor skills [2,23,24]. Normal body posture reflects the even distribution of muscle tone, which guarantees postural balance and goal-directed stepping movement [17,26].

The objective assessment of viscoelastic parameters of muscles was conducted using the Myoton technology, which is well-described in the literature [27], with the MyotonPRO being the latest and more compact version of the device, which is not affected by gravity [28]. Myotonometry is used to describe the viscoelastic parameters of the examined muscle in both healthy patients [20] and patients with neurological deficits [29,30].

A subjective analysis of the patients’ motor patterns showed the impaired quality of support and extension function in both study groups in examination 1. After therapy, in examinations 2 and 3, all children demonstrated improved motor pattern quality, resulting from increased postural control. Similar results of therapy were observed by Eppel et al. (Eppel, 2020) in children with minor postural asymmetry and in adult patients in the early stages after a haemorrhagic stroke. In both groups, the improvement after therapy according to the Vojta approach was defined as significant [31]. Based on their observations, Jung et al. state that the Vojta approach is effective in children with postural asymmetry and conclude that the change in the body posture after this type of therapy is significant [32].

The objective assessment of viscoelastic parameters of the examined muscle, such as frequency, stiffness, elasticity, relaxation, and creep, conducted using the myotonometer during examination 1, carried out prior to therapy, did not reveal any significant differences between the group of children with increased muscle tone and the group of children with non-increased muscle tone. On the other hand, significant differences were demonstrated in examination 2, carried out immediately after the first stimulation according to the Vojta approach. In the group of children diagnosed with increased muscle tone, the recorded frequency and stiffness decreased, and the duration of relaxation increased. Similar observations concerning muscle parameters were described by Nordez et al. (2007). While analysing the changes in the hardness of the plantaris muscle caused by submaximal isometric fatiguing contraction, they observed changes in the viscoelastic parameters immediately after exertion. In the group of children with non-increased muscle tone, no significant changes in the viscoelastic parameters of the muscle were found. Examination 3, conducted after 4 weeks of home therapy, also did not show any differences between the groups in terms of muscle parameters. In both our own research and in the studies conducted by other authors, changes in the postural pattern were observed during the assessment of spontaneous motor skills [15,32,33]. In their research, Goo et al. conducted a study of muscle stiffness in a paediatric group. They carried out a meta-analysis of studies on resting muscle tone and activity in children aged 0–12 years. In terms of neurological development disorders, monitoring the development at the age of 0–2 years is of the greatest importance. As a non-invasive method, myotonometry can be used in patients aged 0–2 years [34]. In their study, Olchowy et al. assessed masseter muscle stiffness in a paediatric population through shear wave elastography using the Aixplorer Ultimate device. Stiffness and other muscle parameters, such as frequency, elasticity, relaxation, and creep in the paediatric population are in the field of interest of researchers who examine a wide range of paediatric issues. Myotonometry performed using the MyotonPro device allows the assessment of five muscle parameters during a single test [35].

The subjective postural analysis of the presented motor patterns in children may lead to indirect conclusions concerning the distribution of muscle tone. The scales most commonly used in paediatrics include the Test of Infant Motor Performance (TIMP) and Alberta Infant Motor Scale (AIMS), and are based on a subjective evaluation [36]. They provide insight into the muscle tone indirectly through the analysis of the postural pattern. These tests do not describe the viscoelastic parameters of muscles or their fluctuations as a result of therapy and over time. None of the analysed methods for determining muscle tone produce objective figures—they only describe the subjective perceptions of the person carrying out the examination. Similar conclusions are presented in the paper by Goo, who analysed the subjective scales for muscle tone assessment in children aged 0–12 months [37].

In this study, the authors observed changes in the duration of relaxation of the erector spinae muscle, a decrease in stiffness and an increase in relaxation after the first implementation of therapy. Similar observations were described by Nordez et al. (2020). Chuang et al. (2012) demonstrated a significant correlation between the increase in muscle length and the increase in its modulus of elasticity. The biochemical properties of muscles, represented by the modulus of elasticity and dynamic stiffness, may decrease as a result of physical activity. The assessment of the viscoelastic parameters of the examined muscle using a myotonometer allows for the isolated quantitative measurement of stiffness of the examined muscular belly. This is relevant from the perspective of objectivisation of the locomotor system diagnostics because it mitigates the external factors that might affect the measurement [30,38,39].

### Limitation

This study was limited by a relatively small number of participants. Changes in the viscoelastic parameters were observed only after the therapy session carried out by a physiotherapist. No changes after the one-month therapy performed by the parents were found. The absence of changes may be attributed to the non-standardised and unsupervised nature of therapy provided by the parents at home.

## 5. Conclusions

Objective diagnostic methods allow the assessment of the viscoelastic parameters of a muscle and the changes occurring in the muscle after therapy. The subjective assessment of the motor pattern is used to classify infants as patients with globally increased or decreased muscle tone. Analysis that includes an objective assessment allows physiotherapists to diagnose local changes in the viscoelastic parameters of the musculofascial structures, which affect the qualitative impairment of the motor pattern.

The fact that the viscoelastic parameters returned to the values obtained during the examination conducted prior to the first therapy session, but at the same time a permanent improvement in the quality of the motor pattern was achieved, may suggest that the local problem that occurred in the muscular belly and blocked the normal development was resolved. The study should be continued with a larger number of participants and with the use of objective measurement tools.

## Figures and Tables

**Figure 1 ijerph-19-10448-f001:**
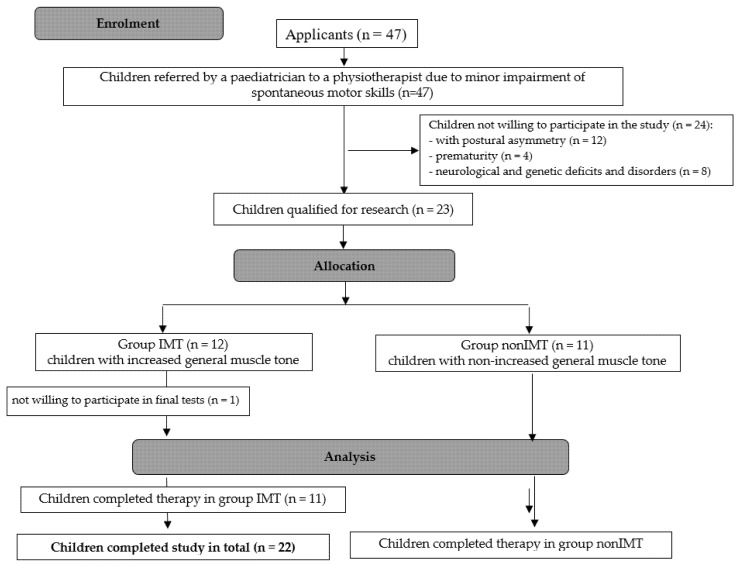
Method for recruiting children to the project.

**Table 1 ijerph-19-10448-t001:** Characteristics of the sample groups.

	IMT Group (*n* = 11)		nonIMT Group (*n* = 11)	
	Mean	SD	Mean	SD
sex (F/M)	3/8		5/6	
age [months]	6.95	3.14	7.05	3.43
Apgar score [points]	9.27	1.10	9.73	0.65

IMT—increased muscle tone, nonIMT—non-increased muscle tone, F—female, M—male.

**Table 2 ijerph-19-10448-t002:** Myotonometric measurement values for the erector spinae in the group of participants with increased muscle tone (IMT) and the group of participants with non-increased muscle tone (nonIMT), obtained from each of the three measurements.

	IMT Group			nonIMT Group		
	Mean	Median	SD	Mean	Median	SD
Frequency 1 [Hz]	21.04	21.90	4.27	22.70	22.80	2.55
Frequency 2 [Hz]	19.98	20.20	2.59	22.55	22.90	2.06
Frequency 3 [Hz]	21.96	22.40	1.94	21.79	22.50	2.42
Elasticity 1	0.99	0.94	0.23	1.09	1.00	0.24
Elasticity 2	1.03	1.02	0.18	0.97	1.04	0.18
Elasticity 3	0.99	0.92	0.12	0.98	1.02	0.24
Stiffness 1 [N/m]	582.70	556.00	135.61	578.36	555.00	87.07
Stiffness 2 [N/m]	460.82	500.00	114.96	582.09	608.00	90.50
Stiffness 3 [N/m]	586.89	592.00	88.86	605.27	546.00	167.07
Relaxation 1 [ms]	8.84	9.05	1.86	9.14	9.10	1.56
Relaxation 2 [ms]	10.95	10.30	1.96	8.98	8.90	1.52
Relaxation 3 [ms]	8.90	9.00	1.46	9.14	9.80	2.16
Creep 1	0.60	0.60	0.15	0.59	0.58	0.09
Creep 2	0.69	0.66	0.11	0.57	0.56	0.09
Creep 3	0.58	0.58	0.09	0.61	0.65	0.14

IMT—increased muscle tone, nonIMT—non-increased muscle tone, 1, 2, 3—examination number.

**Table 3 ijerph-19-10448-t003:** The significance of differences in the analysed parameters between the group of children with increased muscle tone (IMT) and the group of children with non-increased muscle tone (nonIMT), calculated for each measurement (measurements 1, 2, 3).

IMT Group vs. nonIMT Group	Measurement		
	1	2	3
Frequency	0.2810	0.0206 *	0.8707
Stiffness	0.9307	0.0124 *	0.7702
Elasticity	0.3377	0.4538	0.9106
Relaxation	0.6956	0.0182 *	0.7829
Creep	0.7461	0.0123 *	0.5709

IMT—increased muscle tone, nonIMT—non-increased muscle tone, * statistically significant value *p* < 0.05.

**Table 4 ijerph-19-10448-t004:** The significance of the differences between the measurements (*p* ANOVA) and *p*-values for the post hoc Scheffe’s test calculated for the comparison of measurements 1, 2, and 3 in each of the sample groups.

	IMT Group				nonIMT Group			
		Measurement:				Measurement:		
	*p* ANOVA	1 vs. 2	1 vs. 3	2 vs. 3	*p* ANOVA	1 vs. 2	1 vs. 3	2 vs. 3
Frequency	0.0122 *	0.0197 *	0.8922	0.0472 *	0.2087	N	N	N
Stiffness	0.0216 *	0.0356 *	0.9429	0.0657	0.9787	N	N	N
Elasticity	0.4665	N	N	N	0.5441	N	N	N
Relaxation	0.0088 *	0.0172 *	0.9596	0.0292 *	0.4704	N	N	N
Creep	0.1166	N	N	N	0.1118	N	N	N

1, 2, 3—examination number; IMT—increased muscle tone, nonIMT—non-increased muscle tone, N—no statistical significance, * statistically significant *p*-value < 0.05.

## Data Availability

The data presented in this study are available on request from the corresponding author. The data are not publicly available due to a bigger research program.

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
