# Peer review of "Assessment of Viscoelastic Parameters of Muscles in Children Aged 4–9 Months with Minor Qualitative Impairment of the Motor Pattern after Vojta Therapy Implementation"

_ijerph, 2022, doi:10.3390/ijerph191610448_

Round 1

Reviewer 1 Report

 The proposed study addresses the short-term effects of treatment according to Vojta Therapy in children with minor impairment of spontaneous motor skills. The effect on increased muscle tone was assessed by using the MYOTON device.

The interesting finding concerns the improvement of visco-elasticity parameters in the erector muscles of the spine immediately after a therapy session, an effect not observed one month later during which the children underwent parent-administered therapy. The proposed study could be considered a pilot study that does not end with the publication of the preliminary data but paves the way for a larger study with a larger number of sessions and 30- and 60-day follow-up.

In addition, the abstract is misleading by providing information that does not correspond to the results discussed in the paper.

Author Response

Answer to reviewer 1.

Thank you for the time and suggestions. We agree with the review and suggestions. We are convinced that applied changes will improve the quality of our manuscript.

 Below we are answering all suggestions:

The interesting finding concerns the improvement of visco-elasticity parameters in the erector muscles of the spine immediately after a therapy session, an effect not observed one month later during which the children underwent parent-administered therapy. The proposed study could be considered a pilot study that does not end with the publication of the preliminary data but paves the way for a larger study with a larger number of sessions and 30- and 60-day follow-ups. – We are planning to do the larger project with  30- and 60-day follow-ups.  Thank you for that suggestion

In addition, the abstract is misleading by providing information that does not correspond to the results discussed in the paper. – We also change the abstract to be precise about the subject of our research.

Reviewer 2 Report

The study is interesting. However, the study sample is very small and the method is based on a single intervention. The manuscript is worthy of publication after major revision. Due to lack of clarity and poor writing. My comments are below.

Abstract

According to the guidelines abstract should be structured but without headings. Next, the abstract is poorly written. Please add the aim of the study in the background section. Then in the methods add information on what groups were compared and in terms of what. The expression ‘the first study group’ does not carry any information for the reader. Information about the study registration is not needed in the abstract.

Introduction

Overall, the introduction lacks clarity on why the study was conducted, and what novelty researchers wanted to share with the community. Is this the only study of this type?

Page 1, line 35. Information about a referral to the physiotherapist after a diagnosis with a R62.9 code is imprecise. Children with such a diagnosis can be referred to other healthcare professionals as well, e.g. gastroenterologist, orthopaedists. Such diagnosis is not only related to motor development. Please clarify this information.

Methods

Please replace the phrase “sample group” with “study group”. Also, the naming of the study group is unfortunate. Please use other terms than the first and the second group, preferably choosing terms that are related to the purpose of creating such groups.

Page 2, line 83. Please use abbreviations correctly. Each abbreviation should be explained with the first use and then used in the abbreviated form (see IMT, MFDD for example).

Page 2, line 90. In scientific literature, we use Table 1, not Table no. 1. Please correct this throughout the manuscript.

Page 2, line 91. Abbreviations should be listed under tables as specified in the guidelines. Do not use abbreviations in table captions. This applies to all tables in this manuscript.

Page 4, line 133. Figure 1 needs a title.

Table 4. What does N denote? Please explain.

Discussion

Page 6, line 208. Discussion should start with a short summary of the study goal juxtaposed with the key findings.

Page 6, line 209. Why Hbd abbreviation was introduced? If there is no reason, please remove it.

Page 6, line 238. Is Jung a single author of this study? The same with Nordez and other authors later on. If they are not single authors then correct them, please.

Page 7, line 258. Are there any other methods available to evaluate muscle tone? Please write more about them. There are surface electromyography and sonoelastography for example. Shear-wave elastography is particularly interesting. Please refer to the review by Goo et al. for the identification of methods used for the evaluation of the condition of skeletal muscles in children (Goo et al. Ultrasound Med. Biol. 2020, 46, 1831–1840). The example of using elastography in the pediatric population is given by Olchowy et al. Int J Environ Res Public Health. 2021;18(18):9619.

Page 7, line 269. In this next paragraph, the authors write about elasticity but do not name the method of the examination. As mentioned before, elastography is a valid objective method, so this paragraph should be improved and corrected with up-to-date knowledge.

Finally, the language used in this manuscript is poor, in many places lacks clarity. There are a lot of typos, misused articles, and guidelines for scientific writing that are omitted. I pointed out some errors but I recommend that the authors have this manuscript checked by a medical writer.

Author Response

Answer to rewiever 2.

Thank you for revising our manuscript. We are glad that subject of our research is interesting and worth of publication. Kindly thank you for all the comments. We tried to apply the best to all comments.

Please find all our answer to the comments  bellow.

Abstract

According to the guidelines abstract should be structured but without headings. Next, the abstract is poorly written. Please add the aim of the study in the background section. – We enrich abstract by adding aim and more information to clarify what are we were working on.

Then in the methods add information on what groups were compared and in terms of what. The expression ‘the first study group’ does not carry any information for the reader.- Named of group were changed and information were added

 Information about the study registration is not needed in the abstract. – Registration information was transferred to the materials and method paragraph

Introduction

Overall, the introduction lacks clarity on why the study was conducted, and what novelty researchers wanted to share with the community. Is this the only study of this type? – that study is conducted on very young children. We conducted that as a novelty that can be used to objectify the assessment of muscle parameters in the pediatric group.

Page 1, line 35. Information about a referral to the physiotherapist after a diagnosis with a R62.9 code is imprecise. Children with such a diagnosis can be referred to other healthcare professionals as well, e.g. gastroenterologist, orthopedists. Such diagnosis is not only related to motor development. Please clarify this information. – Information was clarified

Methods

Please replace the phrase “sample group” with “study group”. Also, the naming of the study group is unfortunate. Please use other terms than the first and the second group, preferably choosing terms that are related to the purpose of creating such groups. – Sample group was changed to the study group

Page 2, line 83. Please use abbreviations correctly. Each abbreviation should be explained with the first use and then used in the abbreviated form (see IMT, MFDD for example). – Abbreviations were  checked and explained in a correct way

Page 2, line 90. In scientific literature, we use Table 1, not Table no. 1. Please correct this throughout the manuscript. – All table numbers were changed in a correct way.

Page 2, line 91. Abbreviations should be listed under tables as specified in the guidelines. Do not use abbreviations in table captions. This applies to all tables in this manuscript – abbreviations are corrected and replaced

Page 4, line 133. Figure 1 needs a title. -Tittle is added

Table 4. What does N denote? Please explain.- N is explained

Discussion

Page 6, line 208. Discussion should start with a short summary of the study goal juxtaposed with the key findings. – Short summary of the study is added in the second paragraph

Page 6, line 209. Why Hbd abbreviation was introduced? If there is no reason, please remove it. – Abbreviations are removed

Page 6, line 238. Is Jung a single author of this study? The same with Nordez and other authors later on. If they are not single authors then correct them, please.- All the authors were checked and correctly cited

Page 7, line 258. Are there any other methods available to evaluate muscle tone? Please write more about them. There are surface electromyography and sonoelastography for example. Shear-wave elastography is particularly interesting. Please refer to the review by Goo et al. for the identification of methods used for the evaluation of the condition of skeletal muscles in children (Goo et al. Ultrasound Med. Biol. 2020, 46, 1831–1840). The example of using elastography in the pediatric population is given by Olchowy et al. Int J Environ Res Public Health. 2021;18(18):9619.- Information is added and included in the text.

Page 7, line 269. In this next paragraph, the authors write about elasticity but do not name the method of the examination. As mentioned before, elastography is a valid objective method, so this paragraph should be improved and corrected with up-to-date knowledge. – Method is explained in lines 253 -  257 page 7

Finally, the language used in this manuscript is poor, in many places lacks clarity. There are a lot of typos, misused articles, and guidelines for scientific writing that are omitted. I pointed out some errors but I recommend that the authors have this manuscript checked by a medical writer.- Language was corrected by medical translator

Round 2

Reviewer 1 Report

Thank you for paying attention to the requests and editing the text.

Author Response

Answer to reviewer 1 round 2

Thank you for your time.

Reviewer 2 Report

I appreciate that the authors did a considerable effort to improve the quality of the paper. However, it still needs to be polished. Particularly writing and consistency.

As advised before, the paper should be checked for language, preferably by a medical writer. The use of abbreviations is still incorrect. Abbreviations should be defined separately in the abstract and the body of the manuscript. There are some other language errors as well.

Table 1. There is n used for the number of participants I believe. But it is described as a mean, which is incorrect. Please change this in the heading, eg. IMT group (n=11). The current format is not clear.

Figure 1 should be better formatted. The text is partly invisible.

Table 4. Is there any rationale for using numerical p-values for some not significant comparisons and for other ‘N’? Can you please use numerical values everywhere for consistency?

Author Response

Answer to reviewer 2 round 2

Thank you for your time and all directions and advice. We hope we manage to meet the required corrections

Below is our answer.

As advised before, the paper should be checked for language, preferably by a medical writer. The use of abbreviations is still incorrect. Abbreviations should be defined separately in the abstract and the body of the manuscript. There are some other language errors as well the paper was checked by a translator in all suggesting aspects.

Table 1. There is n used for the number of participants I believe. But it is described as a mean, which is incorrect. Please change this in the heading, eg. IMT group (n=11). The current format is not clear. – it is corrected to be clarified.

Figure 1 should be better formatted. The text is partly invisible. – formatting is corrected.

Table 4. Is there any rationale for using numerical p-values for some not significant comparisons and for other ‘N’? Can you please use numerical values everywhere for consistency? Post hoc comparisons were only computed when p ANOVA values ​​showed statistically significant differences (p <0.05). Therefore, for some variables, instead of the p value, there is N (not statistically significant). To make the table more readable and to indicate the reason for the lack of some calculations, the values ​​of the p ANOVA coefficient were added to table 4.
